# Identification of Language-Induced Mental Load from Eye Behaviors in Virtual Reality

**DOI:** 10.3390/s23156667

**Published:** 2023-07-25

**Authors:** Johannes Schirm, Andrés Roberto Gómez-Vargas, Monica Perusquía-Hernández, Richard T. Skarbez, Naoya Isoyama, Hideaki Uchiyama, Kiyoshi Kiyokawa

**Affiliations:** 1Graduate School of Science and Technology, Nara Institute of Science and Technology, Ikoma 630-0192, Japan; gomez_vargas.andres.gz3@is.naist.jp (A.R.G.-V.); m.perusquia@is.naist.jp (M.P.-H.); hideaki.uchiyama@is.naist.jp (H.U.); kiyo@is.naist.jp (K.K.); 2Department of Computer Science and Information Technology, School of Computing, Engineering and Mathematical Sciences, La Trobe University, Melbourne Campus, Melbourne, VIC 3086, Australia; 3Faculty of Social Information Studies, Otsuma Women’s University, Tokyo 102-8357, Japan; isoyama@is.naist.jp

**Keywords:** eye tracking, virtual reality, mental load, cognitive load, flow state, listening comprehension, language task

## Abstract

Experiences of virtual reality (VR) can easily break if the method of evaluating subjective user states is intrusive. Behavioral measures are increasingly used to avoid this problem. One such measure is eye tracking, which recently became more standard in VR and is often used for content-dependent analyses. This research is an endeavor to utilize content-independent eye metrics, such as pupil size and blinks, for identifying mental load in VR users. We generated mental load independently from visuals through auditory stimuli. We also defined and measured a new eye metric, focus offset, which seeks to measure the phenomenon of “staring into the distance” without focusing on a specific surface. In the experiment, VR-experienced participants listened to two native and two foreign language stimuli inside a virtual phone booth. The results show that with increasing mental load, relative pupil size on average increased 0.512 SDs (0.118 mm), with 57% reduced variance. To a lesser extent, mental load led to fewer fixations, less voluntary gazing at distracting content, and a larger focus offset as if looking through surfaces (about 0.343 SDs, 5.10 cm). These results are in agreement with previous studies. Overall, we encourage further research on content-independent eye metrics, and we hope that hardware and algorithms will be developed in the future to further increase tracking stability.

## 1. Introduction

Virtual reality (VR) as a medium is still in development. To assess the performance and quality of VR, there is an increasing need for evaluating the internal state of its users. One common requirement is the estimation of the mental load put on users as a result of how the virtual world is designed [1,2]. The most widely recognized method to estimate mental load is to directly collect ratings from users, either during or after the experience [1,3]. But similar to related user experience factors, the evaluation of mental load is being increasingly supplemented by additional behavioral measures recorded using a variety of sensors. Physiology and behavior quantification can help reduce interruptions of the experience and detect possible issues with the validity of subjective ratings [4,5,6]. The recent spread of high-quality eye-tracking hardware that already comes inbuilt with the head-mounted displays (HMDs), which are used to display VR, is a promising new source of quantitative data for this task [7,8,9].

In VR, eye behaviors are often analyzed at the level of virtual objects. For this, relevant scene objects are identified by the developers of the virtual world, and contacts between the user’s gaze and these objects are logged on a per-object basis. Clay et al. [9] used 2D heat maps and 3D collision points between a user’s gaze and the virtual scene to analyze gaze interactions with the VR content. Asish et al. [10] analyzed how often users moved their gaze away from the central object of a virtual lecture they were asked to pay attention to, with the aim of detecting distraction states during a virtual class. Less research is available on metrics that depend less on the specific contents of the VR scene [11]. We found this to be the case especially for fully stereoscopic VR content consisting of three-dimensional objects that the user can navigate by changing their location and perspective [12]. However, recent work by Callahan–Flintoft et al. [8] and Lamb et al. [7] provided detailed insights into the general availability and peculiarities of eye-tracking data from consumer-oriented HMDs. Both studies agree that the utility of eye tracking depends on the stimulus distance.

A shortcoming of these methods is that they require analyzing the relationship between an object in the VR scene and gaze. An alternative is so-called content-independent methods, which analyze eye behaviors without being constrained by this relationship. Content-independent methods of analyzing eye behaviors in VR may currently be less studied and applied, but the relation between eye behaviors and cognitive states has been identified in many studies [13]. We believe that content-independent eye metrics would ease the detection of cognitive states from eye behaviors and therefore the evaluation of VR. One benefit of this approach is that fewer adjustments to the specific VR contents need to be made, resulting in lower administration costs. However, further investigation is necessary to test the transferability of earlier research carried out using desktop screens [14] to fully virtual, stereoscopic VR. Despite current technical advances, there are still many reasons why transferability between tracking in reality and VR experiences is not straightforward. These include headset weight, limited field of view, artifacts introduced by additional lenses, and the vergence–accommodation conflict [15] (p. 3). This conflict essentially puts the human eye in constant discomfort when wearing an HMD.

Previous work has identified a considerable number of content-independent eye metrics [14]. In the context of this research, we consider three of the most common eye metrics. **Pupil size** is known to increase with mental load [13]. Researchers were able to distinguish several types of breaks in attention only based on pupil size [16]. **Blinks** were found to play a role in blocking out visual input to reduce distraction while thinking. Internal, creative thinking often leads to higher blink rates and longer blink durations than external, analytical thinking [17]. **Eye vergence** refers to the angle between the right and left gaze rays. Also sometimes referred to as “gaze depth,” there has been research on eye vergence in VR [18,19]. Huang et al. [20] found eye vergence to even “outperform existing method[s] using fixations, saccades, and blinks” for detecting moments when viewers of an educational video spaced out. They found that eye vergence tends to fall back into a resting pose when humans disengage with visual input. However, large individual variance and a rapid loss of precision for distant gaze targets make it difficult to take advantage of this eye metric [7,21].

Meta-metrics are derived from basic eye metrics using detection algorithms. We consider the following two meta-metrics. **Fixations** are the times at which gaze velocity and acceleration stay below a predefined threshold. **Saccades** describe the opposite state, during which the gaze wanders around and breaks the threshold. Microsaccades are another major meta-metric, but they require higher sample rates than our VR hardware can currently provide. Previous work suggested that creative/internal thinking produces more and shorter fixations and more saccades than reading text (external thinking) [22]. However, meta-metrics are especially known to more strongly depend on specific tasks or scenarios. One obvious example would be the task of reading a few paragraphs of text—fixations and saccades will almost completely follow the structure of the writing. These data might still inform us about the internal state of the reader, but they need to be put into the context of the specific text, making it a content-dependent measure.

This research has the following aims:The design of an exemplary user task in VR that minimizes bias on eye behaviors as much as possible;The creation of an exemplary VR environment that facilitates the parallel collection of the five aforementioned eye metrics, plus one exemplary content-dependent measure for reference: distraction content;A comparison between eye metrics in terms of their relationship to mental load.

First, we designed a task that allows for the generation of mental load while affecting eye behaviors as little as possible. This is achieved by playing speech in a foreign language to users. In previous research, mental load has been most commonly generated by introducing parallel tasks or time limits, requiring abstract problem solving or motor skills [23] (p. 282). Language comprehension has been studied as the cause of mental load, especially in the field of linguistics [24], but to our knowledge it has not yet been considered as a practical way to generate mental load in the context of media experiences.

Regarding the comparison between eye metrics, we formulated several hypotheses, which are summarized in Figure 1. High mental load would lead to increased pupil size [13], longer and more frequent blinks [17], longer and less frequent fixations, and more intense and less frequent saccades [25]. It would also cause users to more frequently have their gaze fall back into a resting pose [21], as if they were looking through/behind the virtual 3D surface they were facing. Finally, we hypothesized that high mental load would inhibit users from voluntarily looking around. Research on blinks already indicated that humans reduce their engagement with the external world during internal thinking [26]. The intensity of these responses over increasingly difficult conditions was expected to take one of two possible shapes. Either intensity would increase linearly with difficulty, which would more likely be an indicator of physical stress, such as heart rate [6], closer to what subjective ratings tend to reflect [23] (p. 286), or intensity would peak at the optimal ratio of difficulty and skill level for each individual, which would more likely be an indicator of higher activation levels and increased cognitive processing, such as increased pupil size [13]).

In short, mental load is generated naturally or through an explicitly introduced task and leads to specific physiological and behavioral responses. This relationship has been studied in great detail already but is not yet widely taken advantage of in VR. We assess whether such responses can be elicited and captured with current VR hardware and propose a methodology to compare different eye metrics in one combined environment.

The following novelties characterize the proposed methodology:A listening task based on foreign language skills. With only aural cues, eye behaviors can be expected to be independent of visual cues, which are often necessary to generate mental load.The “focus offset” eye metric for VR, calculated from eye vergence and the geometry of virtual surfaces. In other words, the mismatch between “the virtual 3D surface the user is gazing at” and “the actual intersection of the left and right gaze ray”.The design of a coherent virtual scenario. From the early design stages, measures and stimuli were selected with generalizability in mind. A virtual phone booth serves as a narrow space with less than 1 m between the user’s head and the walls. With this, eye vergence values can be expected to be more stable.

With the current research, we make the following contributions.

A protocol using a widely available VR headset. We provide our full procedures and assets for statistically investigating users’ eye behaviors in VR. This is to make our research easy to reproduce, improve, and customize.Listening task based on foreign language skills. In a separate step, we validated a selection of audio stimuli in terms of their impact on perceived mental load. The same procedure can be used with any future combination of stimuli and target groups.Comparison of content-independent eye metrics. An exploratory investigation of the most common eye metrics inside a single scenario provided us with a ranking of eye metrics by explanatory power, in the case of VR.

We expect the eye metrics investigated in this research to be easily applicable to existing VR scenarios. They can be especially useful for use cases such as passive viewing or free exploration, which can be difficult to evaluate using only content-dependent quantitative measures. Focus offset can be helpful in narrow environments but requires the accurate removal of blinks, saccades, and other artifacts. The capabilities of eye-tracking data need to be understood better to allow for appropriate visualization and use in alternative contexts like the Internet of Things [27,28]. In summary, we would like to encourage further research on the utility of content-independent eye metrics in VR evaluation, not only in the context of mental load but also considering important related factors, including its presence in VR, flow, attention, and emotions.

## 2. Base Concepts

For measuring mental load (see definition in Table 1), Gopher [23] recommends combining three approaches: subjective, physiological, and behavioral. The advantages and disadvantages of each approach are listed in Table 2. This research focuses on a combination of physiological and behavioral measures because of their transparency to the user and their accuracy in reflecting physical mechanisms. Still, it is important to be aware of their limits in explaining user intentions and cause-effect relationships.

Subjective measures can provide valuable information on intentions and emotions, but they might not be able to reflect underlying physical mechanisms very well. “Flow processes” [23] (p. 286) exemplify this; by definition (see Table 1) they reduce the meta-awareness needed to accurately assess one’s own experience. Whereas physical mental load—in terms of activation levels [23] (p. 286)—increases when experiencing flow, the person experiencing flow is not aware of it. This can result in dissociations between subjective and physiological/behavioral measures [23] (p. 286). In fact, Peifer et al. found that with linearly increasing stress during a task, heart-rate-measured parasympathetic arousal also increased linearly, but Cortisol-measured sympathetic arousal took the shape of an inverted U [6]. The authors also found this inverted u-shape in subjective ratings of flow.

**Table 1 sensors-23-06667-t001:** Definition, relevance, examples, and relationship of the two main base concepts of this research, “mental load” and “flow experience”.

	Mental Load	Flow Experience
Definition	Construct describing cognitive demands at work or in the household. Historically most widely used in social sciences. Defined as “the cost of mental operations, and the constraints that are imposed by these costs on the ability of a performer to cope with the demands of a task that he or she is given to perform” [23].	Most fundamentally, the space of all possible combinations of task difficulty and skill level as shown in Figure 2. Also influenced by individual, contextual, and cultural factors [29]. High skill and low difficulty lead to boredom, and low skill and high difficulty lead to anxiety. In the center of the space, a linearly rising channel of combinations with approximately equal levels of difficulty and skill emerges.
Relevance	Became an important concept in the field of human–computer interaction with the advent of digital systems and research on their user experience. Also relevant to emerging technologies like VR [2].	Csikszentmihalyi proposed the challenge-skill flow channel [30] (p. 49) already in 1975, but it has become increasingly relevant to fields like game design.
Examples	Researchers most commonly generate mental load by introducing parallel tasks or time limits to base tasks that require users to solve abstract problems or perform motor tasks [23] (p. 282).	Often researched in the context of educational games. Hamari et al. [31] studied two examples of typical tasks eliciting flow: a puzzle-style workbench to solve logical, spatial, and geometric problems, and a navigation-based action game that requires precise physical control. Gamers were also subject to research on links between neural activity and flow [29] (p. 9).
Relationship	A person experiencing flow during an activity located in the flow channel is less likely to experience so-called interludes [30] (p. 38), which are moments of reassessing whether the activity should be continued or mental resources should rather be saved. Consequently, a person in this state is less restrictive when making use of mental resources and therefore more likely to keep taking on higher mental load.	

**Table 2 sensors-23-06667-t002:** Overview of the three major approaches to measuring mental load as discussed by Gopher [23] (p. 286), with an additional row showing which of our measures we classify as which approach.

	Subjective	Physiological	Behavioral
Sensitive to…	Strategic control, voluntary effort, and emotional states	Processing mechanisms, activation levels	Computational costs and structures
Less reflective of…	Computational cost/conflicts, flow processes	Computational conflicts, differences between voluntary/involuntary engagement	Cause of effects (load-related vs. unrelated)
Implementation in this study	Comprehension rating, task load rating	Pupil size	Eye behaviors ^1^, head movements

^1^ Eye behaviors considered in this research: blinks, fixations, saccades, focus offset (see Section 4.1.2), and gaze contact with distraction objects in the periphery of users.

## 3. Experiment 1

There is surprisingly little research on generating mental load through a listening comprehension task, despite attractive benefits like the minimization of task-related visual stimuli. Scientists and language learners increasingly agree that there is a difference between consciously learning a language and acquiring a language [32]. As a generalization of conscious vs. acquired language skills, Gopher already discussed controlled vs. automatic processes and conscious vs. implicit knowledge in the context of mental load [23] (pp. 277–278). He considered limiting generators of mental load to “operations that are under voluntary and conscious control”, but we propose to utilize the interspace between conscious and acquired language skills to control task difficulty. Our proposed task design can be seen in Figure 3.

In this research, we chose to use English and Japanese. The language distance between these two languages has been described as one of the largest observable [33]. English is also a regular part of Japanese education and has been studied for several years by most Japanese. General research on language learning showed that using foreign languages produces different patterns of brain activity compared to using one’s mother tongue [34]. Shinozuka et al. observed different patterns in brain activity between familiar and unfamiliar words of the same language, and between English and Japanese. They suggest this might “reflect the difference in the cognitive loads depending on the levels of automatization in one’s language processing” [35].

For this first experiment, we chose samples of spoken Japanese and English and evaluated the effect of sample difficulty on perceived mental load in Japanese native speakers. Since this experiment primarily sought to evaluate our choice of samples, it was implemented in the form of a web survey.

### 3.1. Methods

#### 3.1.1. Participants

Participants were recruited by email in several universities and on a number of public social media channels. All confirmed that Japanese was their native language and they had not been involved with this research before. Further demographic data have been summarized in Table 3.

#### 3.1.2. Materials and Measures

Four samples of spoken speech with increasing difficulty were chosen. The difficulty level of each sample was determined based on general knowledge about our target group of participants—Japanese university students learning and using English at an (on average) intermediate level. Besides the difference between native (Japanese) and foreign (English) languages, we aimed to differentiate one more level of difficulty within each language, which we will refer to as “familiarity” from now on, following Shinozuka et al. [35]. The contents of the audio samples were as follows:*JP-Familiar*: Spontaneous and enunciated sample in Japanese of a professor introducing the alumni association;*JP-Unfamiliar*: Literary, dense, and more abstract sample from a Japanese movie about a fictional dystopia and morals;*EN-Familiar*: An English sample about today’s air temperature, made for learners of American English;*EN-Unfamiliar*: Spontaneous and fast, abstract sample of a lecture on skill acquisition in American English.

For the full audio samples or references, please see the Appendix A. The audio samples have varying lengths because we wanted to let each speaker finish their thought. *JP-Familiar* was 63 s, *JP-Unfamiliar* was 48 s, *EN-Familiar* was 45 s, and *EN-Unfamiliar* was 47 s long. The average length of sentences—in the sense of coherent statements—was 9.1 s (SD=3.99). The sample rate of the source audio was at least 44.1 KHz. Audacity [36] was used to reduce its frequency range to 200–2000 Hz. This was to simulate telephony audio and prevent the stimuli from being too easy to understand. Using dynamic range compression, the average volume level was normalized to around −12 dB.

Participants were asked to rate their perceived task load after each sample by answering the Japanese version of the NASA-TLX [3,37]. In addition, a custom question item was used as a rating of perceived language comprehension: “How well do you feel that you understood the speaker?” Participants were also asked about their media preferences (“Which media types are you exposed to in everyday life?”) and experience with learning and using English (“How much are you listening to English in everyday life, using textbooks or language learning apps/other material?”). The question about media preferences was implemented as a 4 × 2 grid of checkboxes that could be toggled independently. Each media type was explicitly referred to as domestic (Japanese) media. The eight most common media types in Japan are animation films (anime), video games, novels, comedy, plays, TV dramas, TV news, and radio.

#### 3.1.3. Procedure

In a web-based survey, participants answered questions about demographic data. Then, they listened to the four audio samples in random order. Each audio sample could only be listened to once. After each audio sample, participants answered the NASA-TLX and rated their comprehension. At the end, there was a chance to write an optional free-text comment about the survey experience.

Ethical approval for this study was obtained from the institutional review board at the Nara Institute of Science and Technology (2022-I-25). There was no reward given to the participants of this survey, but it was designed to be very short, lasting only about 6 min. The survey was anonymous.

#### 3.1.4. Statistical Analysis

The following aspects were investigated:Does the order of the audio samples in terms of comprehension difficulty—as perceived by the target group, Japanese university students in this case—match the intended raking *JP-Familiar* < *JP-Unfamiliar* < *EN-Familiar* < *EN-Unfamiliar*?How well does perceived comprehension correlate with the perceived mental load during listening?Looking at perceived mental demand and frustration, which audio sample is most likely to elicit a flow state and therefore a peak in activation levels (see Section 2)?

Furthermore, possible effects of individually different familiarities with certain native language styles were investigated. Posing a comprehension challenge to a person in their native language—which is what *JP-Unfamiliar* was selected for—is inherently difficult as most people are used to an extremely broad spectrum of different speaking styles in their native language. This creates the need to pick a more niche style of speaking, e.g., domain-specific talk or dialects, which in turn gives people an advantage if they have coincidentally been more exposed to this less common style.

### 3.2. Results

Figure 4 shows an overview of all reported NASA-TLX scores. Spearman’s rank correlation was computed to assess the relationship between the “mental demand” scale of the raw TLX scores and perceived comprehension scores. There was a statistically significant strong, negative correlation between these two variables (ρ(118)=−0.73, p<0.001), which is also visible in Figure 5.

To confirm that the audio samples are increasing in difficulty, three (paired) tests were carried out, one for each increase in difficulty between adjacent audio samples. The three *p* values from these tests have been adjusted using the Benjamini–Hochberg procedure. Cohen’s effect size is reported as *d*.

Ratings of perceived mental load during *JP-Familiar* and *JP-Unfamiliar* were taken from the raw TLX scores. A Shapiro–Wilk test indicated that the distribution of these values deviated significantly from normality (W=0.85, p<0.001). A Wilcoxon test was carried out to evaluate whether perceived mental load differed between *JP-Familiar* and *JP-Unfamiliar*. A significant difference (p<0.01, d=0.48) was found, showing an increase in perceived mental load during *JP-Unfamiliar* (M=23.16, SD=26.36) compared to *JP-Familiar* (M=12.28, SD=18.09).

Ratings of perceived mental load during *JP-Unfamiliar* and *EN-Familiar* were taken from the raw TLX scores. A Shapiro–Wilk test indicated that the distribution of these values deviated significantly from normality (W=0.90, p<0.001). A Wilcoxon test was carried out to evaluate whether perceived mental load differed between *JP-Unfamiliar* and *EN-Familiar*. A significant difference (p<0.05, d=0.37) was found, showing an increase in perceived mental load during *EN-Familiar* (M=32.63, SD=24.31) compared to *JP-Unfamiliar* (M=23.16, SD=26.36).

Ratings of perceived mental load during *EN-Familiar* and *EN-Unfamiliar* were taken from the raw TLX scores. A Shapiro–Wilk test indicated that the distribution of these values was normal (W=0.98, p=0.638). A Levene test indicated homogeneity of variances (F(1,58)=1.32, p=0.255). There was a significant increase in perceived mental load during *EN-Unfamiliar* (M=71.23, SD=20.58) compared to *EN-Familiar* (M=32.63, SD=24.31), t(29)=−8.96, p<0.001, d=1.71.

Next, frustration ratings were taken from the raw TLX scores. A Shapiro–Wilk test indicated that the distribution of these values deviated significantly from normality (W=0.89, p<0.001). A Kruskal–Wallis rank sum test was carried out on frustration by audio sample. A statistically significant difference was found ($\chi^{2}\left(3\right)=24.82$, p<0.001). A pairwise Wilcoxon rank sum tests with Benjamini–Hochberg *p* value adjustment were carried out. *EN-Unfamiliar* (M=42.46, SD=23.08) was significantly different from all other audio samples, *JP-Familiar* (M=15.44, SD=22.28, p<0.001, d=1.19), *JP-Unfamiliar* (M=20.35, SD=27.70, p<0.01, d=0.87), and *EN-Familiar* (M=15.79, SD=19.69, p<0.001, d=1.24), in that it featured a much higher frustration mean. Finally, Figure 6 gives an overview of possible influences of media exposure on perceived mental load.

### 3.3. Discussion

The survey results confirmed a significant negative correlation between perceived mental load and perceived comprehension. This is the expected pattern of increasing mental load as comprehension decreases and more active thinking becomes necessary. In Figure 5, the raw TLX value starts rising relatively early, making it an overall more sensitive measure. However, the correlation of the two measures is sufficient (ρ=−0.73).

Significant increases in perceived mental load between adjacent audio samples provided evidence for a steadily increasing difficulty level over all four audio samples. Moreover, *EN-Unfamiliar* was found to have caused significantly higher frustration levels than all other audio samples. This suggests that in the case of our audio samples and target group, the peak of physical activation levels (see Section 2) can be expected to be located before *EN-Unfamiliar*.

Figure 6 revealed a slight anomaly for *JP-Unfamiliar*, where consumers of video games, anime, and novels appeared to report less mental load. This makes sense, as *JP-Unfamiliar* is an excerpt from a science fiction anime. However, while there might be individuals with more exposure to this literary style of speech, it was overall still perceived as more difficult than *JP-Familiar* and easier than *EN-Familiar*.

In summary, the four selected audio samples from Section 3.1.2 were found to produce increasing perceived mental load in our target group of Japanese participants. The experiment in Section 4.1 will contain the same four audio samples to produce mental load during a VR experience.

## 4. Experiment 2

The second experiment was to collect eye-tracking data from users of a VR scene that integrated the four audio samples validated in the first experiment described in Section 3.

Using standard features of the game engine Unity [38] and two free assets from its asset store [39,40], we built a virtual phone booth as an environment for listening to the audio stimuli. Figure 7 shows the full scene and a close-up of the phone with some posters on the side as potential distractions. Participants were only instructed to understand the contents of the speech samples and could look around freely if they desired. The green glowing sphere in a glass bulb on top of the phone was pulsing slightly as the amplitude of the currently played audio changed. This was to give the audio more presence in the virtual world while at the same time provide a default fixation point as an “opposing force” to the distraction posters, allowing for the detection of voluntary exploration. In agreement with Kern et al. [41], the audio was spatialized to the position of the green sphere using the Steam Audio Spatializer [42]. Spatial Blend was adjusted to a ratio of 0.8. A ratio of 1.0 yielded an extreme fall-off when turning only one ear to the sound source, resulting in the other ear receiving silence.

The scene was displayed on the HTC Vive Pro Eye VR headset (Valve Corporation, Bellevue, WA, USA) and rendered by Unity 2019.4.13f1 (LTS) on Windows 10 (1909) with an Intel Core i7-10700, GeForce GTX 1660 (6 GB) and 16 GB of RAM (PC4/DDR4). The initial standpoint of participants was on average 2.63 m (SD=0.26) in front of the phone booth. In the phone booth, the space to stand in was a little more than 1 m diameter. The entrance was 1.95 m high, the green sphere was 1.5 m above the ground, and the posters were distributed between 1.2 m and 1.4 m of height. An invisible, centered collider was used to detect when participants moved their head too far outside the phone booth. This was to make sure their head position was close enough to the audio source at all times.

While also being a natural environment for listening to phone calls, the phone booth provides a narrow space within the recommended 2–3 m range for stable eye vergence [18]. The closest point between the left and right gaze ray is one of the main parameters for focus offset, but its position becomes extremely sensitive to noise with increasing distance. (The two gaze rays rapidly approach being parallel to each other as users look more into the distance). Furthermore, the interior of the phone booth was lit more brightly compared to the outside. This was to avoid large pupil sizes due to low light, which can decrease eye tracking precision [43]. Two virtual light sources were set: a spotlight for subtle lighting from the outside, and a point light with an intensity of 0.65 at the ceiling of the phone booth.

### 4.1. Methods

#### 4.1.1. Participants

Fourteen males and one female with a mean age of 22.87 (SD=0.92) participated in our experiment. All were native speakers of Japanese. They rated their current motivation and actual efforts to improve at foreign languages on a 5-point Likert scale (with 1 being “I don’t like language learning” and 5 being “I spend time every day”) on average to be 3.5 (SD=0.9).

Most participants were students of the same research group but had not been informed about the contents or objective of the experiment beforehand. All participants gave their informed consent for inclusion before they participated in the study. The study was conducted in accordance with the Declaration of Helsinki.

#### 4.1.2. Materials and Measures

The subjective measure for this experiment was a 5-point Likert item “How well do you feel that you understood speaker number *x*?” with candidates from 1 “Not at all” to 5 “Very good”. This item is based on the perceived comprehension rating from the survey in Section 3.1.3 and was shown to correlate well with the NASA-TLX scores for perceived mental load (ρ=−0.73, see Section 3.2). It was answered for all presented speech samples in one go as part of the post-questionnaire. Reducing subjective measures to one item helped participants in accurately answering the post-questionnaire as they had to remember their experience with each of the four samples and their order.

The main dependent variable was eye-tracking data obtained from the inbuilt eye tracker of the HTC Vive Pro Eye VR headset, which has a precision of about 3°. This approximate precision changes with head movement and gaze distance from the view center [44]. The SDK’s sensitivity level was set to 0.8 to smooth the otherwise noisy data by inbuilt algorithms. The following values were contained in the raw data:**Head transform**: The headset position in world-space, view/up vector;**Gaze points**: The closest collision point with a virtual 3D surface for left and right gaze ray (world-space);**Focus point**: The point at which the left and the right gaze ray came closest to each other (world-space);**Pupil size**: The pupil diameter in millimeters, −1 for closed eyes or errors;**Gaze target**: The name of the gazed-at object (e.g., the name of distraction poster).

From these raw data, the **duration and frequency of blinks**, **fixations**, and **saccades** were detected. In addition, **focus offset** was calculated as a VR-derivative of eye vergence. We define it as the offset of the “real focus point (intersection of gaze rays)” from the “closest point on the surface currently gazed at”. To reduce noise for more distant surfaces, we divide this offset by the surface distance, as given by the distance between the user’s head and the point on the surface they are gazing towards. To code the direction of the determined offset, we define values behind the surface to be positive. Consequently, negative values are those that are closer to the user. Figure 8 shows the following three major cases: (a) zero when focusing exactly on the surface, (b) positive when looking “through” the surface, and (c) negative. Negative values would imply that the person at least temporarily squinted a bit, but it can also hint at noise and calibration errors.

#### 4.1.3. Procedure

To keep the conditions the same between participants, they were only told how to wear the VR headset and proceed through the displayed instructions using their gaze. The program then automatically explained the task, performed the necessary calibration for eye tracking, gave the opportunity to adjust the audio volume, and recorded some eye-tracking data for verification. The verification data were recorded to allow for the detection of potential calibration issues in the analysis. For this, participants were asked to follow a crosshair with their gaze, which moved on a predetermined path within their field of vision. For details on the preparation steps and the specific wording used to instruct participants, please refer to the Appendix A. A gray overlay covered the virtual environment during the whole duration of the preparation phase.

After the preparations were finished, the gray overlay vanished and the phone started ringing. To make participants adjust to the VR scene, they were first placed outside the phone booth and had to enter it by taking 3–4 steps forward. As members of the same research group, participants were familiar with the physical environment and the VR headset. The experimenter further made sure that no cables or other objects were in the way, and since the rest of the VR experience only required standing in a fixed location, the setup did not pose any dangers to participants. The four audio samples (see Section 3.1.2) were then played back in random order with a few seconds delay and a phone beep at the beginning. In case a participant moved outside the collider cube (see Section 4), the audio was stopped and the phone started ringing again. The current audio sample was then restarted once the person moved inside the collider again. This was to keep participants from moving away too far from the phone, which can create problems with eye tracking and also reduces audio volume unnecessarily.

After the experience ended, participants took off the headset and immediately answered the post-questionnaire at a PC in the same room. For the full post-questionnaire, please refer to the Appendix A. The whole procedure was performed in a room without other persons apart from the experimenter. The VR experience lasted on average 4.15 min (SD=0.67). Preparing and reading the instructions usually took no longer than 3 min. The post-questionnaire could be completed in about 6 min, although some participants took more time to write about their general impressions. Most participants were able to finish the complete procedure within 15–20 min and were then compensated with candy and snacks.

#### 4.1.4. Statistical Analysis

First, we visualized all recorded data in a compact way and searched for obvious problems, such as participants inadvertently closing their eyes. We also evaluated how accurately each participant was able to follow the crosshair during the verification explained in Section 4.1.3. After manual inspection, we removed four male participants because of problems with parts of their eye-tracking data. Only the last full playback of the audio sample was used in three trials where the participant moved too far out of the phone booth.

Next, the raw eye-tracking data needed to be cleaned. Blink detection was performed using GazeR by Geller et al. [45] (p. 2244). The commonly used 100 ms extension before and after blinks was increased to 200 ms because of the lower sample rate. To compensate for the variation in sample rate, the function noise_based_blink_detection of the GazeR package was modified to return sample indices instead of times. Data during blinks was linearly interpolated whenever possible, including 3D points. The function moving_average_pupil was used to smooth the pupil size of each eye separately.

Finally, pupil size was baseline-corrected per audio sample (trial) using the data between 500 ms and 2000 ms as a reference. This is a longer span than is commonly used, but due to the general length of each trial, we found this time window to be appropriate. Left and right pupil sizes were then combined into one representative pupil size value based on their sample-wise mean. This is also common practice in other literature and can compensate for eye-specific noise or calibration issues [45].

Focus offset (see Section 4.1.2) and head/gaze vector rotations were calculated sample-wise. Due to strong noise in the focus offset, it was smoothed using a Savitzky–Golay filter with a window size of 201. The window size was selected upon visual inspection of the results. Smoothed focus offset still featured visible individual variance, as shown in Figure 9. To make the focus offset comparable, the values were then normalized by centering them around their median for each participant.

Despite theoretical advances in detecting meta-metrics like fixations and saccades in VR [11], we have not yet been able to find a non-proprietary tool for this task. Therefore, we mapped the 3D gaze information onto a virtual plane in front of the user and made use of an existing algorithm. While the relation between gaze direction and head movements gets lost with this approach, the algorithm we adopted has the advantage that it is adaptive and can more stably deal with individual differences in gaze behavior [46].

The conversion was performed by intersecting the user’s combined 3D gaze ray going from their head to the intersection of their left and right gaze ray. For this purpose, a virtual plane was created at 3 m distance in the user’s head-space. Virtual screen size was then determined by centering all 2D intersection points around their median and only considering points that were not extreme outliers in the 0.0025 quantile of all four edges. With the dimensions calculated, the coordinates were transformed into a standard screen coordinate system. Pixel coordinates were then calculated using a virtual resolution of 150 dpi. The result was a virtual screen with a custom size per trial, depending on each person’s eye movement activity.

With this, gazepath by Renswoude et al. [46] was able to detect fixations and saccades. However, it assumed a uniform sample rate, which made it necessary to use the time information of the first and last sample of each listening phase to correctly scale the generated event times to the actual length of the listening phase. When mapping the results back onto the samples this way, event times were rounded inwards to avoid overlaps. Very short events of just 1–3 samples in length may have been dropped during this process.

For the statistical analysis, listening phases were split into sections of 10 s each. This value is based on the average sentence length in our set of audio samples (see Section 3.1.2) and includes ≈0.5 s of buffer at the beginning and end of each section. To make conditions equal for comparison, only the first four sections were used. With the audio sample lengths stated in Section 3.1.2, 40 s in total would include most of the data while being slightly shorter than the shortest audio sample. After summarizing the data of each of the four sections (mean or number/length of certain events), this resulted in n=44 per audio sample group.

### 4.2. Results

#### 4.2.1. Post-Questionnaire

Their ratings for perceived comprehension (on a 5-point Likert scale from 1 for worst to 5 for best) were on average 4.9 (SD=0.3) for *JP-Familiar*, 3.1 (SD=1.5) for *JP-Unfamiliar*, 3.6 (SD=0.9) for *EN-Familiar*, and 1.6 (SD=0.7) for *EN-Unfamiliar*.

Ratings for perceived comprehension were taken. A Shapiro–Wilk test indicated that the distribution of these values deviated significantly from normality (W=0.95, p<0.01). A Kruskal–Wallis rank sum test was carried out on perceived comprehension by audio sample. A statistically significant difference was found ($\chi^{2}\left(3\right)=39.01$, p<0.001). Pairwise Wilcoxon rank sum tests with Benjamini–Hochberg *p* value adjustment were carried out. With *d* being Cohen’s effect size, significant differences were found between *JP-Unfamiliar* and *EN-Unfamiliar* (p<0.01, d=1.30), and for all other comparisons (p<0.001, d>1.9). No significant difference was found between *JP-Unfamiliar* and *EN-Familiar* (p=0.151).

#### 4.2.2. Eye Behaviors

The center of this research is the eye-tracking data described in Section 4.1.2. We noticed that the sample rate was not uniform because the eye-tracking data were taken at the game engine’s frame rate. On average, data were recorded with a rate of 90.26 Hz (SD=5.81). We were able to compensate for this in later steps.

While listening, head position was on average (−0.52,−3.34) cm (SD=(3.13,8.21)) on the x-z-plane (top view), which is within 10–15 cm of distance to the center of the phone booth. With an average of 0.5–1 m distance between eyes and surface, focus offset was expected to be stable most of the time [18] (p. 7).

##### Content-Independent

Table 4 summarizes all statistical analyses performed on content-independent eye behaviors and if applicable, references the corresponding plot in Figure 10.

For each eye metric, a Shapiro–Wilk test (to detect value distributions deviating significantly from normality) and a Levene test (to detect violations of the homogeneity of variances in the values) were performed. A violation of at least one of these conditions was detected for all eye metrics. Following this, a Kruskal–Wallis rank sum test was performed on the whole set of values to determine the presence of any significant differences between audio sample groups. If a significant difference was found, pairwise Wilcoxon rank sum tests with Benjamini–Hochberg *p* value adjustment were performed between all combinations of audio sample conditions. Effect sizes between the values of significantly different audio sample conditions were calculated using Cohen’s *d*.

Overall, the results in Table 4 show that 5/12 eye metrics detected a significant pattern: pupil size means, pupil size variances, focus offset means, number of fixations, and head movement variances. In addition, tendencies toward the same pattern of eye metrics with a significant change can be seen in the plots of focus offset variances (Figure 10d), fixation duration means (Figure 10e), and number of saccades (Figure 10g). Especially for fast-paced events like blinks and saccades, no change in the corresponding eye metric could be identified.

##### Content-Dependent

A distraction ratio was calculated for each of the three distraction posters as described in Figure 11. This calculation was only performed for the combinations of participants and audio samples for which eye contact with any of the three posters happened at least once, which was the case for 22 out of 44 combinations in total. A statistical test was performed to confirm the visually identified pattern of *JP-Unfamiliar* and *EN-Familiar* exhibiting lower distraction ratios.

A Shapiro–Wilk test indicated that the distribution of the calculated values deviated significantly from normality (W=0.85, p=0.003). A Kruskal–Wallis test was carried out on participant-wise distraction ratios of looked-at posters, by audio sample. A statistically significant difference was found ($\chi^{2}\left(3\right)=8.47$, p<0.05). Pairwise Wilcoxon rank sum tests with Benjamini–Hochberg *p* value adjustment were carried out. For *JP-Unfamiliar* (M=0.12, SD=0.07), *JP-Familiar* (M=0.46, SD=0.33, p<0.05, and d=1.23), and *EN-Unfamiliar* (M=0.36, SD=0.16, and p<0.05, d=2.13) were significantly different and featured higher distraction ratios. This means that for participants who actually got distracted by the posters, the distraction “share” of *JP-Unfamiliar* as compared to *JP-Familiar* and *EN-Unfamiliar* was, on average, 34% and 24% lower, respectively.

### 4.3. Discussion

The four audio samples from Section 3.1.2 were selected to provide increasing difficulty, and a separate online survey confirmed increasing levels of perceived mental load in Japanese native speakers for this set of audio samples (see Section 3.1.4). However, according to the perceived comprehension ratings given by our VR participants, they did not find *EN-Familiar* significantly more difficult than *JP-Unfamiliar* (see Section 4.2.1). The means suggest that this is not due to *EN-Familiar* being particularly easy but rather due to *JP-Unfamiliar* being unexpectedly difficult. Reasons for this could be more modest ratings by some VR participants or the smaller *n* in comparison to the online survey from Section 3.1.4. There could also have been fewer participants this time who happened to be already more familiar with the style of speech in *JP-Unfamiliar*.

Nevertheless, many eye metrics indicated a significant response between audio samples, and almost all of the metrics displayed a trend towards the same effect. In fact, all of the metrics except head rotation featured a response peak at *EN-Familiar*, the second most difficult sample. This matches the inverted u-shape physiological indicators of physical activation, which can accomodate increasing cognitive demands, a pattern that was also identified in flow research (see Section 2). These findings are further supported by the online survey, where high frustration levels during *EN-Unfamiliar* were detected, indicating that the peak of the inverted u-shape in question is most likely located before *EN-Unfamiliar* (see Section 3.2).

Large differences were found in pupil size means, all towards *EN-Familiar*, in which pupil size had increased the most. A significant difference was also found in pupil size variances between *EN-Familiar* and *JP-Familiar*, with *EN-Familiar* displaying much less variance. Interestingly, pupil size variance was lower for the condition with higher mental load (*EN-Familiar*) in this case. Previous research observed the opposite, with “transient” increases in pupil size leading to higher variability [47]. There is a chance that the language task achieved more sustained attention than other tasks, but this would need to be addressed in a separate experiment comparing different task types.

Focus offset means also featured a significant difference, although less clearly related to *EN-Familiar*. Especially considering the high dispersion within *JP-Familiar* (and a little bit in *EN-Unfamiliar*, see Figure 10c), we think that despite our efforts to isolate only fixations to calculate focus offset, too much gaze activity (looking around) introduced noise to this metric. However, the same pattern as in pupil size means (the most distinct eye metric) can be observed for clips other than *JP-Familiar*. We suspect that an imperfect counterbalancing made participants look around more in *JP-Familiar* than in other clips, which created more focus offset noise. For focus offset variances, there was only a slight trend of *JP-Unfamiliar* having less variance in the visual comparison (see Figure 10d). Generally, our analysis of focus offset confirms strong individual variance, which had been discussed in similar studies on eye vergence [21].

The work by Salvi et al. [17] led us to expect longer and more frequent blinks, especially because this was found to be a more deeply trained physiological mechanism to shut out visual input while thinking. However, no significant differences in blink duration or frequency were found. We think this is due to the comparatively low sample rate.

Fixation frequency was found to have significantly dropped during *EN-Familiar* compared to *JP-Unfamiliar*. For mean fixation durations, a trend towards longer fixations during *EN-Familiar* could only be observed visually (see Figure 10e). Again, we think that the weaker effect on durations is likely due to the comparatively low sample rate.

Saccade frequency only showed a visual trend towards fewer saccades during *EN-Familiar* (see Figure 10g). For saccade durations, there is almost no visible difference between audio samples. Since saccades are very short-lived events of a few hundred milliseconds, this measure might also require a higher sample rate.

A new pattern was found in head movement variances, where the only significant difference was found between *JP-Familiar* and *EN-Unfamiliar*, the easiest and the most difficult audio sample. One way to interpret this is that although many participants did not enter a flow state and were therefore not displaying physical signs of cognitive activity while listening to *EN-Unfamiliar*, they continued to wait for comprehensible passages while keeping their heads still. For gaze movement variances, no differences between audio samples could be observed.

The distraction posters (see Section 4) seemed to have been effective. Although the “distraction ratio” (see content-dependent metrics in Section 4.2.2) was only significantly different in *JP-Unfamiliar* compared to *JP-Familiar* and *EN-Unfamiliar*, a similar trend can be seen for *EN-Familiar* in Figure 11. The ratio was only calculated for listening phases during which the participant gazed at a poster at least once. Because of this, the comparison was performed with a small number of samples (n=22), and a single outlier for the right poster in *EN-Familiar* had a great impact. The results suggest that a higher mental load made it less likely for participants to voluntarily look around. The higher distraction ratios for *EN-Unfamiliar* seemingly contradict our findings on head movement, but we think the two metrics capture two different types of individual behavior: (1) most participants trying to silently wait for a comprehensible passage in *EN-Unfamiliar*; and (2) some more easily distracted participants giving up listening and starting to explore the environment. The distraction ratio only captures the extent of (2) as it can only be calculated for participants who were distracted at least once.

There are several limitations to our study. eye-tracking data were sampled at only around 90 Hz instead of the 120 Hz technically available with the hardware used due to the way data recording was implemented. We could mostly compensate for this during pre-processing but suspect that fast-lived events like saccades and blinks were not fully captured. We also converted 3D gaze information to be compatible with an existing algorithm for detecting screen-space fixations and saccades, but to account for the vestibulo-ocular reflex, a VR-optimized algorithm might be preferable [7].

The randomization of the four audio samples was not ideal. A check of the random distribution of the four audio samples unfortunately showed that their order had been biased. If 1 means “always first audio sample” and 4 means “always fourth audio sample,” average positions were 1.73 (SD=1.10) for *JP-Familiar*, 2.73 (SD=1.35) for *JP-Unfamiliar*, 2.64 (SD=0.67) for *EN-Familiar*, and 2.91 (SD=1.04) for *EN-Unfamiliar*. All positions would ideally approach 2.5, but *JP-Familiar* had the highest chance of coming at the very beginning. At this time, participants had just entered the scene and were most likely to take a first look around the environment. We suspect that our new focus offset metric was affected by this as its calculation is most sensitive to noise induced by saccades.

We aimed to design the virtual environment and the listening task in a way that the results would be transferable to the most common types of virtual scenes. However, future research has yet to evaluate potential dependencies of the eye metrics presented in this research, for example, on the narrow space or the passive viewing situation.

## 5. Conclusions and Future Directions

In this study, native speakers of Japanese walked into a virtual phone booth and listened to audio samples at four levels of difficulty. We expected them to display typical eye behaviors in response to increased mental load while comprehending more difficult speech. We further introduced focus offset as a VR-compatible metric describing how far users look through the virtual surface they are facing and expected its value to increase with higher mental load.

Using the inbuilt eye tracker of the HTC Vive Pro Eye, we confirmed effects on pupil size (mean and variance), focus offset (mean), and fixations (frequency). We observed similar tendencies in focus offset (variance), fixations (duration), and saccades (frequency). The strongest response occurred on the second to most difficult audio sample, *EN-Familiar*, instead of the most difficult audio sample, *EN-Unfamiliar*. It appears that in the case of our participants, *EN-Familiar* provided the right level of difficulty without creating too much frustration. This is in agreement with studies from flow research (see Section 2), which also observed this type of inverted u-shape in physiological measures during increasingly demanding tasks. Head movement variance was the only metric where the most difficult audio sample indicated most change (decrease). We interpreted this as reflecting the willingness of most participants to keep trying to understand despite the felt frustration during this audio sample. Distraction posters outside the main area of interest were less likely to be looked at for audio samples that corresponded to the peak of the inverted u-shape.

In short, the right balance of challenge and skill leads to actual mental load (in the sense of physical arousal), which again leads to certain physiological and behavioral responses. Current VR hardware is accurate enough to detect most important indicators used in related work. The main challenges for future work are the necessary increase in sample rate for the precise detection of blinks and saccades, and the development of methods to stabilize eye vergence without relying on specific VR content.

The proposed listening task can be mostly orthogonal to an existing VR scenario, especially if it does not make strong use of audio, in particular speech. Depending on the language profile of users, it should even be possible to combine the listening task with existing voice contents by switching to another language. The difficulty level can be changed by varying dialects, domains of speech, language distance, or rare/unique speaking styles. However, the selection of appropriate stimuli requires knowledge of each user’s language profile and should be tested in a broader study beforehand, for example, using a survey similar to the one described in Section 3.1.4. Future work could also investigate the effect of parallel, task-relevant visual stimuli, which are not covered by this research. In particular, it would be helpful to understand how attention is redistributed between the audio, task-relevant visuals and the distracting visuals for voluntary viewing.

On the technical side, we strongly recommend future researchers to configure their devices for the maximum sample rate. Especially in the case of games and VR, it is a common mistake to record eye-tracking data at the update rate of the game engine. However, this update rate is (1) not static and (2) is bound to the display rate of the screen or VR headset used, making it impossible to take advantage of (potentially available) higher sample rates. To do this, a separate execution thread and sometimes even a completely different device API becomes necessary.

The second experiment in VR (see Section 4) had comparatively few participants (n=15), and only one of them was female. Further studies are necessary with large populations and better gender balance to confirm the findings of the present study.

In summary, we think that the utility of content-independent eye metrics for quantitatively evaluating user states in VR should be further explored. The focus offset feature would become more useful if data models were created to guide the calculation of eye vergence similar to Orlosky et al. [48]. This time, we focused on statistically identifying mental load between predefined conditions, but in the future, a real-time approach similar to Vortmann et al. [49] could provide great value for interactive VR evaluation. Whenever feasible, experimenters can additionally take advantage of colliders outside the main area of interest as an indicator of distraction.

## Figures and Tables

**Figure 1 sensors-23-06667-f001:**
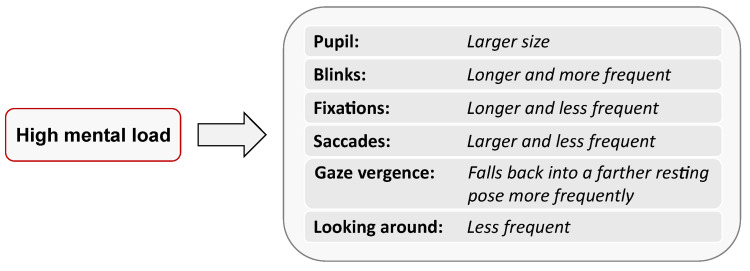
Summary of the hypotheses formulated for this research regarding eye metrics variation under high mental load.

**Figure 2 sensors-23-06667-f002:**
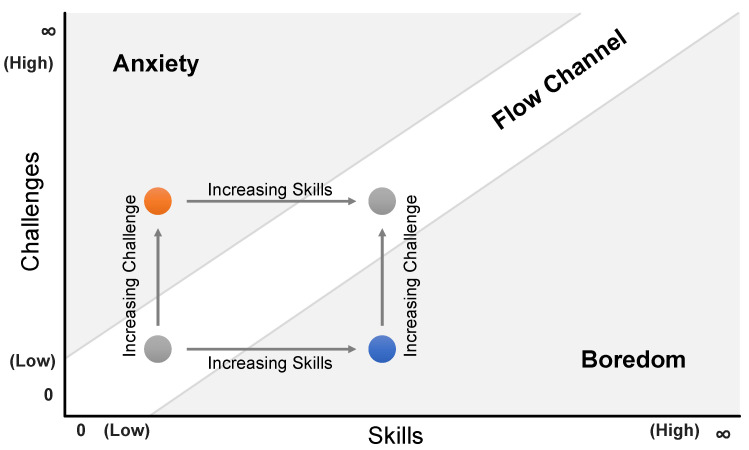
Challenge-skill flow channel proposed by Csikszentmihalyi [30] (p. 49). While a high level of difficulty combined with a low level of skills leads to anxiety and frustration (see orange dot), a low level of difficulty when there is a high level of skills leads to boredom (see blue dot). Flow occurs when challenge and skills balance out.

**Figure 3 sensors-23-06667-f003:**
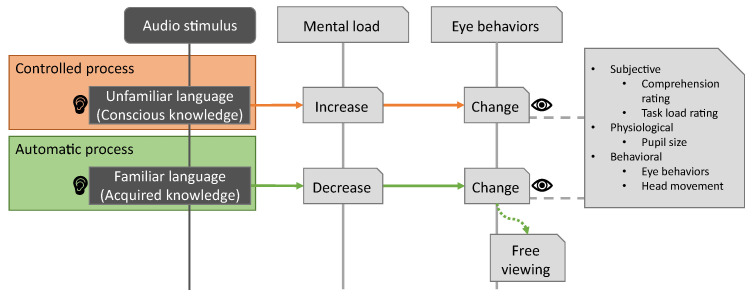
The proposed language task, visualized over time as a sequence diagram. Unfamiliar spoken language increases the mental load of a person trying to understand it. This tends to be reflected in eye behaviors. Familiar spoken language can be understood with almost no additional mental load. This tends to be reflected in eye behaviors too and might even invite users to freely view parts of the scene that are not relevant to the task.

**Figure 4 sensors-23-06667-f004:**
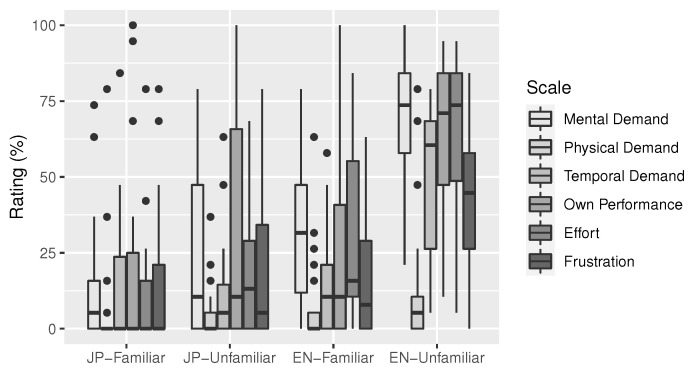
Overview of the raw TLX scores (perceived task load [3]), grouped by audio sample. Each of the six colored items assesses one aspect of the perceived task load while listening to each of the four audio samples. Perceived mental demand (item 1) steadily rose with difficulty. Frustration (last item) suddenly rises for the most difficult audio sample *EN-Unfamiliar*, which reduces the likelihood that participants entered a flow state when listening to it. Black dots represent outliers.

**Figure 5 sensors-23-06667-f005:**
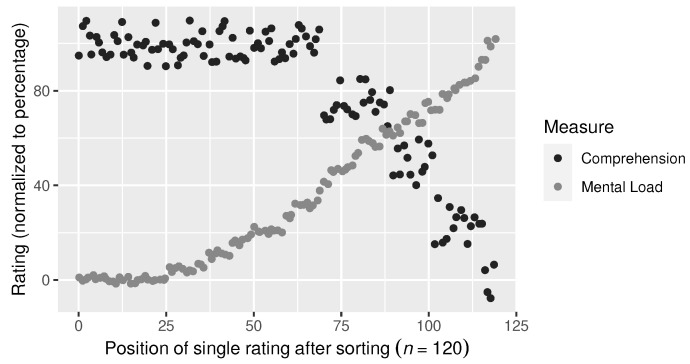
Visualization of the overall correlation between the 5-point Likert scale for perceived comprehension and the raw TLX item for perceived mental demand. The scale for comprehension (ranging from 1 to 5) was mapped to percent. Also, the ratings for comprehension were sorted descendingly to visualize their negative correlation to mental demand.

**Figure 6 sensors-23-06667-f006:**
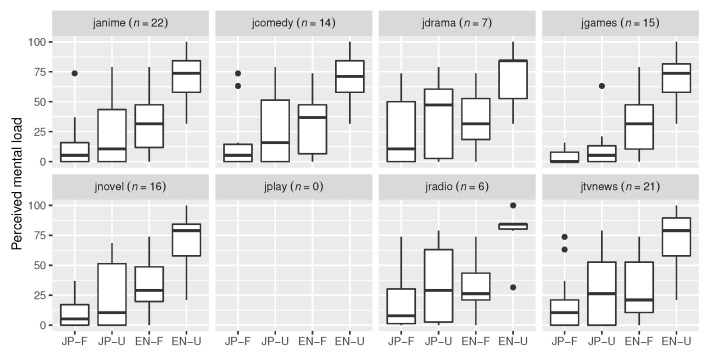
Raw TLX scores for the perceived mental demand of participants consuming each of the media types. Participants could consume multiple media types if they desired. Audio samples labeled “F” are familiar, while samples labeled “U” are unfamiliar. Groups by media types featured mostly similar patterns. For *JP-Unfamiliar*, the differently-shaped distribution of the games group and the low medians of the anime/novels groups stand out.

**Figure 7 sensors-23-06667-f007:**
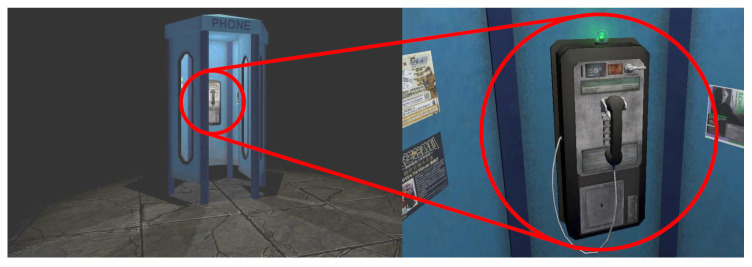
The VR scene only contains a phone booth. A slightly pulsing sphere on top of the phone visualizes played audio. Some posters in Japanese and English are put next to the phone.

**Figure 8 sensors-23-06667-f008:**
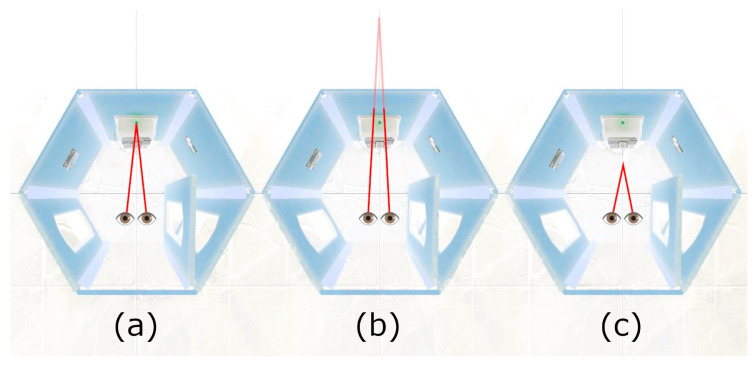
Example of focusing on the green light in the phone booth. For illustration purposes, the red lines are assumed to be parallel to the floor. (**a**) Normal focus on the surface; focus offset mostly zero. (**b**) Hypothesis that higher mental load will cause users to look “through” the surface more often; focus offset is positive. (**c**) Individual behavior or tracking errors may lead to negative values.

**Figure 9 sensors-23-06667-f009:**
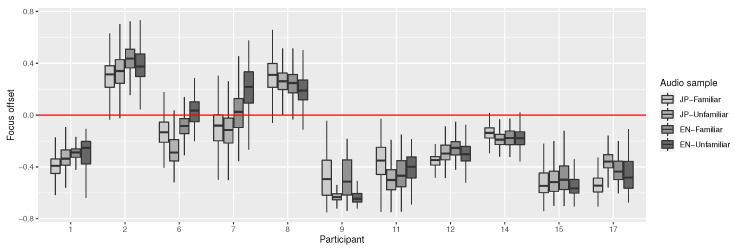
Focus offsets for each audio sample, per participant. Zero (red line) is surface level, and positive values represent a focus offset behind the surface. Data during blinks or saccades have been removed in pre-processing. Outliers were hidden due to their large numbers, for readability.

**Figure 10 sensors-23-06667-f010:**
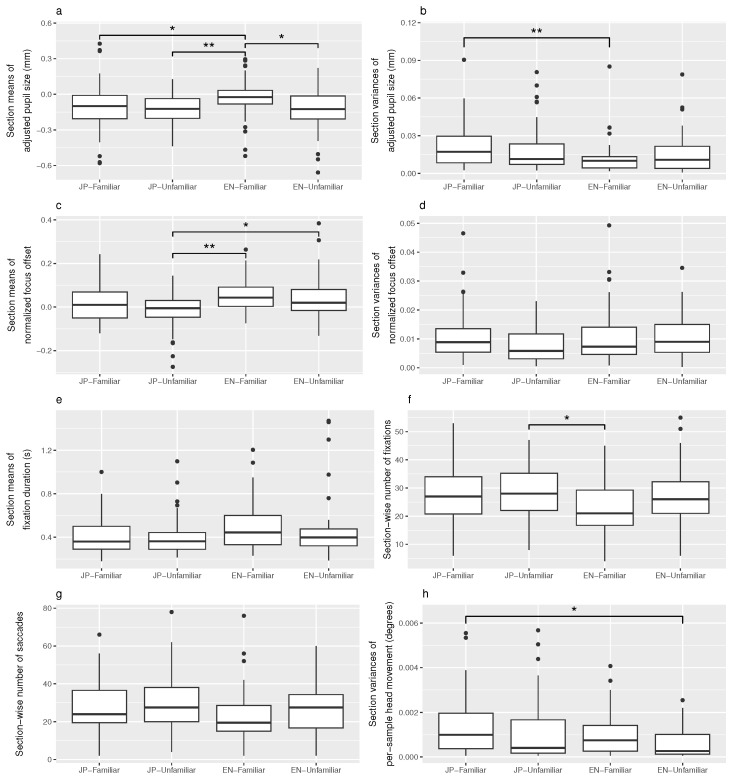
Section-wise aggregates of eye-tracking data by audio sample. (**a**) shows pupil size means. (**b**) shows pupil size variances, with four removed outliers from *JP-Familiar* and *EN-Unfamiliar*. (**c**) shows focus offset means. (**d**) shows focus offset variances, with seven outliers removed mainly from *JP-Familiar*. (**e**) shows fixation durations, with one outlier removed from *JP-Familiar* and *EN-Familiar* each. (**f**) shows fixation counts. (**g**) shows saccade counts. (**h**) shows head movement variability, with thirty-five outliers removed. All outlier removals were performed to allow for a better view of the rest of the data.
* p<0.05, ** p<0.01.

**Figure 11 sensors-23-06667-f011:**
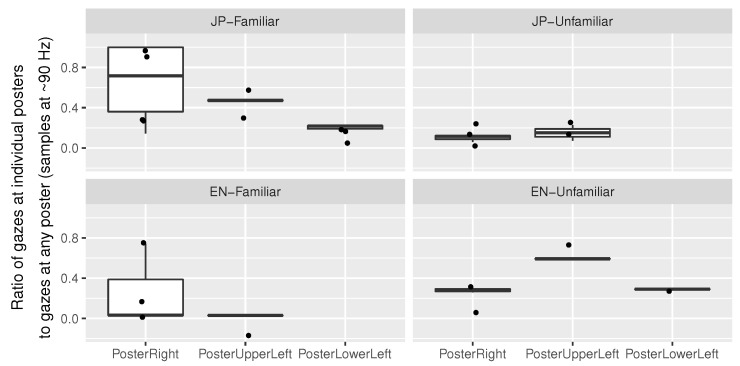
Ratios of the “distraction impact” of the three distraction posters. Calculated by dividing “the number of samples that a participant looked at each poster” by “the number of samples that the same participant spent looking at any of the three posters”. Only 7 out of the 11 participants with usable eye-tracking data looked at the posters at least once.

**Table 3 sensors-23-06667-t003:** Participant demographics, as collected through an online survey.

*n*	Male	21
	Female	9
Age (years)		27.63(SD=7.33)
Efforts to learn English ^1^		1.6(SD=1.1)
Opportunities to use English ^2^		3.5(SD=1.7)
Consumption of Japanese media	Anime	73%(22)
	TV news	70%(21)
	Novels	53%(16)
	Games	50%(15)
	Comedy	47%(14)
	TV Dramas	23%(7)
	Radio	20%(6)
	Plays	0%(0)

^1^ Current efforts to learn English using teaching materials on a 5-point scale (1 being “almost none” and 5 being “more than three times a week”). ^2^ Current opportunities to use English in everyday life on a 5-point scale (1 being “almost none” and 5 being “more than three times a week”).

**Table 4 sensors-23-06667-t004:** Statistical analyses of section-wise aggregates of eye-tracking data, grouped by audio sample. In this research, time series data were aggregated by sections of exactly 10 s (see Section 4.1.4) prior to statistical analysis. Aggregation functions were mean, variance, and number of detected meta-metric events.

Eye Metric (Section-Wise)	Plot	Shapiro–Wilk Test *W*	Kruskal–Wallis Rank Sum Test $\chi^{2}\left(3\right)$	Pairwise Wilcoxon Rank Sum Test with Benjamini–Hochberg *p* Value Adjustment						
				Audio 1			Audio 2			
				Audio	*M*	*SD*	Audio	*M*	*SD*	*d*
Pupil size ^1^ mean	Figure 10a	0.97 ***	12.44 **	*EN-Familiar*	−0.024	0.153	*JP-Familiar*	−0.097	0.180	0.38 *
							*JP-Unfamiliar*	−0.119	0.173	0.66 **
							*EN-Unfamiliar*	−0.208	0.179	0.6 *
Pupil size ^1^ variance	Figure 10b	0.63 ***	10.16 *	*EN-Familiar*	0.013	0.014	*JP-Familiar*	0.030	0.043	0.55 **
Focus offset mean	Figure 10c	0.98 *	14.33 **	*JP-Unfamiliar*	−0.020	0.084	*EN-Familiar*	0.050	0.076	0.88 **
							*EN-Unfamiliar*	0.046	0.098	0.72 *
Focus offset variance ^2^	Figure 10d		7.56							
Blink duration mean		0.82 ***	1.26							
Number of blinks		0.98 *	1.50							
Fixation duration mean	Figure 10e	0.73 ***	7.09							
Number of fixations	Figure 10f	0.98 *	9.00 *	*EN-Familiar*	23.11	10.00	*JP-Unfamiliar*	28.34	9.64	0.53 *
Saccade duration mean		0.49 ***	3.40							
Number of saccades	Figure 10g	0.96 ***	5.25							
Head movement variance ^3^	Figure 10h	0.41 ***	8.14 *	*JP-Familiar*	0.017	0.038	*EN-Unfamiliar*	0.008	0.018	0.3 *
Eye movement variance ^4^		0.23 ***	4.37							

* p<0.05, ** p<0.01, and *** p<0.001. ^1^ Pupil sizes during listening phases spanned from 3.25 mm to 7.74 mm, with a mean of 5.51 mm (SD=0.93). Note that these analyses were performed on baseline-corrected pupil sizes (see Section 4.1.4). ^2^ A Levene test indicated that the homogeneity of variances was violated in the resulting values (F(3,172)=3.56, p=0.015). ^3^ For head movement, the variances in sample-wise rotation angles of the headset’s view vector per section were taken. ^4^ For eye movement, the variance in sample-wise rotation angles of averaged left and right gaze rays per section were taken.

## Data Availability

Supplementary Materials, source code, and aggregated data for this work can be found on the following Open Science Framework repository: https://osf.io/rny6k/. The full Unity project is available on GitHub: https://github.com/jojosoft/language-booth-vr (accessed on 13 July 2023).

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
