# Peer review of "Identification of Language-Induced Mental Load from Eye Behaviors in Virtual Reality"

_sensors, 2023, doi:10.3390/s23156667_

Round 1

Reviewer 1 Report

The authors present novel metrics to assess mental load in virtual reality, offering promising avenues for future research. However, due to the simultaneous impact of auditory and visual stimuli on attention, the experimental setup hindered obtaining conclusive outcomes. Although the posters included fixation points to encourage voluntary exploration, it would have been beneficial for the authors to delve deeper into the influence of attention on performance in dual tasks, such as attention redistribution. Consequently, the connection between head movement and gazing at distraction posters remains ambiguous. Additionally, the limited sample rate employed in the study precluded obtaining statistically reliable results.

Reviewer 2 Report

Thank you for the opportunity to review this paper. I hope these comments are useful in improving the manuscript.

INTRODUCTION

A very well written introduction, that is clear and concise and guides the reader well through pertinent literature. 

Section 2.2, on the flow-experience could benefit from more integration and/or contrast with traditional conceptualisation of workload. Similarly, some practical examples from either the work-environment or gaming could be added.

The paper would definitely benefit from a more traditional structure I would recommend re-structuring he paper as a typical 2 experiment paper – first the validation of the workload measure, and second the VR experiment. This would make it much more clear for the reader.

EXPERIMENT 1 - METHODS

Section 3 should be a traditional methods section (labelled as such) and adopt the structure of:

Participants

Materials and Measures

Procedure

Statistical Analysis

The current section 3.1 (language task) is perhaps too descriptive and could be made more concise. The introduction could include a brief overview of language tasks in terms of workload, such that the method section could just refer to the specific task, rather than the theoretical background.

EXPERIMENT 1 - RESULTS

Section 3.3 introduces the results -but this should be labelled as its own section. Otherwise – well presented.

EXPERIMENT 2 (currently labelled 4 Virtual Environment, 5 Experiment and 6 Results)

There is some “bleed” between the sections, with a statement on statistical analysis as the first sentence of the results (instead of being in the methods) and description of likert scales in the results as well. The results should just present the results. There is also presentation of results in the discussion (see line 668-671 and line 663 as examples). The results should be presented in the results section and the discussion should discuss these results in the broader theoretical context and extant literature. 

Overall the methods section could be clearer and more concise.

FIGURES

Box plots are presented in different formats throughout – these should be standardised.

Well written.

Reviewer 3 Report

Overall, this is a good research. The technological advance to me is quite nice and aimed to utilize content-independent eye metrics, such as pupil size and blinks, for identifying mental load in VR users. The results show that with increasing mental load, mean and variance of pupil size increased. But on the other hand, the paper requires minor adjustments on the following comments:

1.      Mention the contributions of the study in the last paragraph of the introduction section

2.      Explained the novelty of the methodology?

3.      In section 2 the paragraphs are very lengthy please shorten it and need to include recently published works such as

·       https://doi.org/10.22937/IJCSNS.2023.23.2.14

·       https://doi.org/10.21833/ijaas.2021.01.013

Reviewer 4 Report

1. The description of the results in the abstract should be more detailed, adding some data or numerical descriptions of changes in indicators, such as the rate of change in pupil size under different language stimuli and how much focus has shifted.

2. Paragraphs 4, 5, and 6 can provide a general description. The language needs to be simplified to outline the relationship between the eye indicators to be used and psychological burden.

3. Table1 is misplaced and should not be placed in the introduction. It should be placed below the description of the position of table1.

4. It is recommended to delete the paragraph 132-139.

5. The content of section 111-131 can be merged to briefly summarize the problems existing in existing research. The research purpose and methods used in this study can be summarized and summarized, without too much content.

6. Part 2. Base Concepts, merge them into a paragraph, and then draw a table to explain and describe the two terms' mental load 'and' flow experience ', and describe the relationship between them.

7. Explain the colors represented by the circles in Figure 2.

8. Place Figure 6 below the position mentioned in the article.

9. In section 3.3.2. Participants of the article, a table is provided to provide readers with clearer and clearer guidance on the participants.

10. Section 3. Language Task, create a comprehensive technical flowchart that covers a wide range of topics in the article.

11. What does bold font on line 376 mean?

12. It would be better to merge Chapters 3 and 4.

13. What does the bold font in Figure 10 mean?

14. There are too many subheadings in Part 6, so the statistical analysis results are presented in a table to describe factors such as Eye Behaviors and Post questionnaire. A result graph can be described in two paragraphs. There are still many areas in the article that require such modifications. The result description section contains a large number of statements such as "A Kruskal Wallis test", which is a bit repetitive. For example, 6.23 to 6.28 are the results of Figure 9, which can be described in a subheading. A Kruskal Wallis test can be made into a table for overall statistical summary description.

15. The discussion section is well written, but it is recommended to merge Chapter 7 Discussion and Chapter 8 Limitation for description. For the problem of insufficient sample size leading to unclear experimental results, some solutions or future improvement plans can be proposed.

16. Figure 7 can draw three diagrams, the main view, top view, and left view, to describe the example of focusing on the green light of a telephone booth from different perspectives, as only the top view cannot show the difference in height.

17. The introduction and conclusion sections of the article can provide a supplementary description of the relationship between mental load, eye behaviors, and VR.

English language is fine  with minor editing. 

Round 2

Reviewer 1 Report

The article has been much improved. The text below the figures could be made more concise. Additionally, certain figures exhibit low contrast between the test and the background.

Reviewer 2 Report

Thank you for your revisions to the manuscript. I feel the authors have sufficiently improved the manuscript to warrant consideration for publication. 

Reviewer 3 Report

Dear Authors

I requested to add following citation also and it is relevent to the study in creating the patern.

·       https://doi.org/10.3991/ijoe.v17i13.20167

Rest of the comments are addressed well.

N/A for  me..
